# Recent Advances in Multifunctional Antimicrobial Peptides as Immunomodulatory and Anticancer Therapy: Chromogranin A-Derived Peptides and Dermaseptins as Endogenous versus Exogenous Actors

**DOI:** 10.3390/pharmaceutics14102014

**Published:** 2022-09-22

**Authors:** Francesco Scavello, Mohamed Amiche, Jean-Eric Ghia

**Affiliations:** 1IRCCS Humanitas Research Hospital, 20089 Rozzano, MI, Italy; 2Laboratoire de Biogenèse des Signaux Peptidiques (BioSiPe), Institut de Biologie Paris-Seine, Sorbonne Université-CNRS, 75252 Paris, France; 3Department of Immunology, Rady Faculty of Health Sciences, University of Manitoba, Winnipeg, MB R3E 0T5, Canada; 4Section of Gastroenterology, Department of Internal Medicine, Rady Faculty of Health Sciences, University of Manitoba, Winnipeg, MB R3E 0T5, Canada

**Keywords:** antimicrobial peptides, multifunctional antimicrobial peptides, chromogranin A-derived peptides, immunomodulators, dermaseptins, anticancer activities

## Abstract

Antimicrobial peptides (AMPs) are produced by all living organisms exhibiting antimicrobial activities and representing the first line of innate defense against pathogens. In this context, AMPs are suggested as an alternative to classical antibiotics. However, several researchers reported their involvement in different processes defining them as Multifunctional AMPs (MF-AMPs). Interestingly, these agents act as the endogenous responses of the human organism against several dangerous stimuli. Still, they are identified in other organisms and evaluated for their anticancer therapy. Chromogranin A (CgA) is a glyco-phosphoprotein discovered for the first time in the adrenal medulla but also produced in several cells. CgA can generate different derived AMPs influencing numerous physiological processes. Dermaseptins (DRSs) are a family of α-helical-shaped polycationic peptides isolated from the skin secretions of several leaf frogs from the *Phyllomedusidae* family. Several DRSs were identified as AMPs and, until now, more than 65 DRSs have been classified. Recently, these exogenous molecules were characterized for their anticancer activity. In this review, we summarize the role of these two classes of MF-AMPs as an example of endogenous molecules for CgA-derived peptides, able to modulate inflammation but also as exogenous molecules for DRSs, exerting anticancer activities.

## 1. Introduction

The first line of response of mammalians against pathogenic invasion is innate immunity [1]. It consists of different molecules produced and released by various cell types belonging to the organs, immune or other systems [1]. Among these molecules, we find proteins with direct antimicrobial activity or those which activate the complement proteins [1]. Furthermore, small peptides play an essential role thanks to the presence of specific cationic sequences that interact with the membrane of the pathogens [1,2]. These molecules are called host defense peptides and are more widely known as antimicrobial peptides (AMPs) [2]. They can not only act through different antimicrobial mechanisms of action but [1,2] also with different intensities against an extensive collection of pathogens [1,2,3,4]. However, in light of the potential human clinical application, they can be classified as endogenous AMPs produced by the human organism, such as Defensins, Cathelicidins and Dermcidins [3,4]. On the other hand, exogenous AMPs are produced by microorganisms themselves, plants, insects, amphibians and fishes or mammals but not identified in humans, such as Thionins, Piscidins, Cecropins and Dermaseptins [5,6,7,8,9]. However, increasing data show that these AMPs are multifunctional peptides (MF-AMPs) with different roles in cardiovascular, nervous and renal systems [10,11,12]. They are also reported as chemokines, vaccine adjuvants, regulators of the innate defense, immunomodulators and anticancer agents [13,14]. All these properties of MF-AMPs confer to these molecules a broad spectrum of potential applications in the biomedical field. The 20th century will be remembered as the time when antibiotic resistance became a world health problem [15]. Different studies reported this issue and recommended limiting the use of antibiotics to contain the evolution of resistant bacteria [15,16]. However, to the present day, the problem is not solved. The World Health Assembly recognized antibiotic resistance as an alarming issue in human medicine and a leading cause of worldwide death [15,17]. In this context, the MF-AMPs appear to be a possible alternative to antibiotics and have acquired an increasingly clinical interest [18]. At the same time, prevention of nosocomial infection and inflammatory activation of the intervention site after the implantation of medical devices is a significant hospital challenge [19]. Additionally, in this case, MF-AMPs are promising, and several studies reported their functionalization of biomaterials, conferring antimicrobial and anti-inflammatory properties [20]. Finally, some of these agents can modulate systemic and tissue inflammation and immune cell activation [21], supporting future clinical applications as immunomodulators. At the same time, several studies showed the activities of these agents against different models of cancer cells, demonstrating a profile of MF-AMPs with potential clinical application also in cancer therapy [14]. Therefore, the principal aim of this review is to report AMPs as multifunctional agents for future clinical applications. More specifically, we will focus on endogenous Chromogranin A-derived peptides for immunomodulation and Dermaseptins as exogenous agents for cancer therapy.

## 2. General Features, Mechanism of Action and Possible Clinical Application of MF-AMPs

AMPs are effective agents for killing or blocking the growth of free microorganisms due to the self-endogenous origins and natural factors produced by the organisms [4,22]. In addition, the endogenous AMPs are not toxic at high concentrations, and some of these are reported as multifunctional agents with immunomodulatory activity without generating pathogen resistance [22]. Among these, cationic AMPs target bacterial cell membranes in a non-specific manner for pathogens. Still, they are not toxic for host cells due to the specific electrostatic interaction with microbial membrane compounds, such as negatively charged lipids or specific microbial ligands for antimicrobial peptides [23,24]. These agents belong to the host defense response of vertebrates’ innate immune system [23,24]. More than 2000 AMPs are known to be derived from several organisms and catalogued in the AMPs Database [25,26,27]. The peptides are classified based on four different biochemical characteristics: 1. net charge (anionic, cationic and neutral); 2. length of the primary structure (more or less than 100 amino acids); 3. typology of secondary or tertiary structure (linear with β-sheet, α-helix, extended loop and very complex structure, such as cyclic peptides); and 4. hydrophobicity (amphipathic, hydrophobic and hydrophilic) [23,28,29] (Figure 1).

These molecules’ general mechanism of action consists of an initial phase with membrane targeting based on electrostatic interactions with lipids and membrane ligands [23,28]. After achieving a threshold concentration, the molecular interactions between the peptides and membrane induce a conformational phase transition of these molecules. This effect is characterized by a modification from the disordered structure of peptides, typical in aqueous environments, to α-helical or β-sheet conformation upon interaction with phospholipids in the pathogen membrane [23,28]. Then, the peptides can destabilize the membrane inducing electrostatic tension. Finally, membrane disintegration is obtained by pore formation with micellization or lipid segregation (Figure 1). This phenomenon is caused by different molecular mechanisms, such as self-association and multimerization, barrel-stave mechanism, toroid pore or wormhole mechanism and carpet mechanism [23,24,28]. In addition, due to their lipophilic profile, some AMPs do not degrade the membrane but may translocate across it. These peptides exert a cell-penetrating peptides (CPPs)-like function. They induce inhibition of intracellular functions by blocking the cytoskeleton growth, causing membrane dysfunction inside the cell or inhibiting extracellular biopolymers and DNA/RNA/protein synthesis [22,23,28] (Figure 1). In recent years, these agents have acquired an increasing interest in clinical applications. They may represent a novel and promising therapeutic alternative to conventional antibiotics for preventing and eradicating resistant pathogens. AMPs also prevent pathogens’ biofilm formation due to antimicrobial activity [20,29]. In addition, AMPs are elective to destabilize existing biofilms targeting matrix proteins or signaling pathways for growth and essential metabolic processes or compounds [20,29]. In these aspects, another therapeutic utilization of AMPs is preventing infection in the surgical sites and microbial biofilms on biomaterials often associated with the onset of nosocomial infections. Different biomaterials functionalized with peptides and peptidomimetics agents are available for clinical applications. As a few examples, in addition to medically implanted devices, they can be used orally and for wound sites in the surgical area [30,31]. Other vital interests have been reported for endogenous MF-AMPs based on their properties to modulate systemic and tissue inflammation and immune cell activation [21], supporting future clinical applications as immunomodulators. On the other hand, numerously exogenous MF-AMPs are reported as anti-proliferative against several cancer cells with potential clinical application in cancer therapy [14].

## 3. CgA-Derived Peptides as Inflammatory Modulator Molecules

### 3.1. CgA-Derived AMPs

Chromogranin A (CgA) is a glycoprotein with 431 residues and a molecular weight of 49 kDa belonging to the Granin family, discovered for the first time at the end of the 1960s in the granules of adrenomedullary chromaffin cells [32]. In the last few decades, CgA has been identified in immune cells [33,34,35], neurons [36], cardiomyocytes [37], keratinocytes and fibroblasts [38]. This protein may influence different physiological processes. Its role was reported in cardiac function and cardio-protection [39], catecholamine storage and feedback release [40] and the modulation of vascular function [41] but also in cellular recruitment and the modulation of immune response [32,35,42]. However, this prohormone produces, by proteolytic processing, active biological peptides, such as Vasostatins (Vs), Prochromacin, Chromacin, Pancreastatin, WE 14, Catestatin (Cts), Parastatin and Serpinin [43,44]. At the end of the 1990s and early 2000s, several CgA-derived peptides were discovered as AMPs acting against several bacteria, fungi and yeast. The CgA-derived peptides have been found in biological fluids involved in host-defense responses, such as serum, saliva and neutrophils secretions, or against pathogens in the first barrier of the human body, such as the skin [33,34,38,45,46]. The CgA-derived peptides act as antimicrobial agents in the micro-molar range [33,46,47,48]. These concentrations are also reported in the biological fluid after stimulation with pathogen toxins or during infection [33,34,42,46]. Among the CgA-derived peptides, the Vs-I and Cts were first identified as antimicrobial agents; however, their antimicrobial domains were rapidly reported. Vs-I was initially identified as a vasoinhibitory agent [49]. For Vs-I, Lugardon characterized this peptide’s antimicrobial activities against many pathogens [33,47,50]. However, after the incubation of Vs-I with endoproteinase Glu-C, a digested sequence CgA47-66, called Chromofungin (Chr), was identified and found highly active against several fungi and yeasts [47]. It has a global hydrophobicity and amphipathic character, allowing a strong interaction with the membrane. Specifically, Chr possesses a positive charge of +3.5, showing an amphipathic helix in the C-terminal part in the sequence CgA53-66 and at the N-terminal domain, a hydrophobic sequence corresponding to CgA48-51 and a hydrophilic structure CgA53-46, respectively [47,51]. The Vs-I and Chr antimicrobial mechanism of action is explained through the specific interaction of peptides with ergosterol, one of the main components of yeast and fungal membranes, inducing increased pressure and penetration into the membrane [47,51] (Figure 1). Other data demonstrated that Chr could inhibit Calcineurin activity by interacting with Calmodulin [47] (Figure 1). Within microbial cells, Vs-I and Chr may interfere with the Calcium/Calmodulin/Calcineurin signaling pathway by blocking the pathway implicated in virulence and skeleton development of cell walls [52]. Cts was identified as a catecholamine release-inhibitory peptide [53]. Cts is a small 21-amino-acid cationic peptide with a positive net charge of +5 within the bovine sequence (bCgA344-364) possessing a C-terminal hydrophobic sequence. Taylor and colleagues identified a smaller peptide (CgA344-358) derived from Cts with a more substantial inhibitory effect on catecholamine release [54]. This peptide was called Cateslytin (Ctl) by Briolat et al. and is also characterized by its antimicrobial activities with potent effects compared to Cts [46]. Ctl is also a positively charged (+5) arginine-rich antimicrobial peptide and, in an aqueous solution, is a linear peptide with a disordered structure. However, when interacting with the membrane, Ctl acquires an α-helical form [55]. Other studies with a system mimicking bacterial membrane demonstrated that Ctl could convert its structure into antiparallel β-sheets precipitating against the negatively charged part of the membranes [56]. Then, Ctl induced an increased rigidity, permeability gradient and membrane pore formation in the domains containing ergosterol [56,57,58] (Figure 1).

### 3.2. CgA-Derived Peptides and Immune Cells Activities and Inflammation

The involvement of CgA and its derived peptides in innate immunity is well known for its antimicrobial activity. In addition, the role of these MF-AMPs is also reported in immune cells, conferring them a complex profile of immunity modulators. This section analyzes this profile studied using in vitro and in vivo models. The role of CgA-derived peptides on immune cells was studied for the first time in 2009; in particular, the effects of Chr and Cts were evaluated on polymorphonuclear neutrophils. After the treatments with the peptides, Chr and Cts were observed inside the cells, demonstrating their ability to penetrate the mammalian membrane and the profile of CPPs [34]. Then, in the presence of extracellular calcium, the two peptides induced a transient calcium influx in the cells, binding Calmodulin-binding factors (W7 and CMZ) and activating iPLA2 [34]. In addition, the pharmacological block of these channels inhibited the calcium flux induced by Chr and Cts [34]. On the other hand, when extracellular calcium is absent, the peptides cannot induce calcium secretion [34]. Notably, the secretion of polymorphonuclear neutrophils treated with the two CgA-derived peptides induced the secretion of several important factors for innate immunity and inflammation, such as Lactotransferrin, Lysozyme, Neutrophil Gelatinase Associated Lipocalin and S100 calcium-binding protein A8/A9 [34]. Several studies reported the role of Vs-1 as an anti-atherogenesis and anti-inflammatory factor suppressing the adhesion of monocytes to endothelial cells by adhesion molecule down-regulation [59,60]. Xiong and coworkers reported the anti-inflammatory role of Vs-II (N-terminal fragment of CgA containing Vs-I; CgA1-113) in an apolipoprotein E-deficient (ApoE^−/−^) mice model fed with a high-fat diet developing atherosclerosis. In this study, Vs-II treatment reduced the occurrence of atherosclerotic plaque and attenuated lesions [59]. Furthermore, Vs-II significantly reduced the production of pro-inflammatory cytokines in aortic tissue, such as Tumor Necrosis Factor-α (TNF-α), Monocyte Chemoattractant Protein-1 (MCP-1) and Vascular Cell Adhesion Molecule-1 (VCAM-1) [59]. The same authors demonstrated, by several in vivo analyses, that these anti-inflammatory properties are based on the ability of Vs-II to reduce leukocytes adhesion on ApoE^−/−^ mice arteries but also on the recruitment, transmigration and accumulation of M1 macrophages in the lesions [59]. In the same animal model, Sato et al. showed that Vs-I treatment reduces aortic atherosclerotic lesions development due to reductions in intra-plaque inflammation, macrophage infiltration and aortic smooth muscle cells proliferation and plasma glucose level [60]. From a cellular point of view, Vs-I suppressed the lipopolysaccharide (LPS)-induced production of chemokine MCP1 and vascular damage markers, such as VCAM-1 and E-selectin, in human endothelial cells [60]. At the same time, Vs-I was found to reduce M1 pro-inflammatory macrophages differentiation and IL6 release but also oxidized low-density lipoprotein (oxLDL)-induced foam cell formation of macrophages [60]. Of great clinical interest, Vs-I is expressed around Monckeberg’s medial calcific sclerosis in human radial arteries [60]. Additionally, the immunomodulatory role of Chr has been reported in a mice model of ulcerative colitis induced by dextran sulfate sodium administration [61,62]. This model decreased Chr expression [62]. Furthermore, Chr treatment, by intracolonic administration, significantly reduced the inflammation and severity of colitis. The anti-inflammatory effects were due to the differentiation of macrophages into M2 anti-inflammatory clones with the consequent reduction of released IL-18 and the increased expression of M2 markers [61,62]. Using the same animal model, Kapoor and colleagues demonstrated that intrarectal Chr treatment reduced colitis severity and inflammation [63]. In parallel, this was associated with a significant decrease in the expression of CD11c, CD40, CD80, CD86 IL6 and IL12p40 in the inflamed colonic mucosa, mesenteric lymph nodes and spleen [63]. In addition, Chr reduces in CD11c positive cells the expression of CD80, CD86 and NF-κB in the spleen and colon, respectively [63]. All these in vivo data demonstrated that Chr has protective properties against intestinal inflammation and exerts the role of immunomodulator for intestinal macrophages and dendritic cells (DCs). These effects were also demonstrated in vitro with macrophages showing that Chr increased the production of anti-inflammatory factors and the M2 differentiation [61]. At the same time, Chr treatment significantly reduced the expression of M1 macrophage markers and the activation of the NF-κB pathway [62]. Additionally, in this case, in vitro experiments with M1 macrophages demonstrated that this peptide could decrease cellular migration, proinflammatory cytokines production and release, and NF-κB phosphorylation [62]. Furthermore, treatment with Chr or a conditioned medium of Chr-treated macrophages M2 induced epithelial cell proliferation and migration but also decreased oxidative stress and pro-inflammatory cytokine production [61]. In addition, the Chr treatment of naïve bone marrow-derived CD11c positive DCs reduced the LPS-induced expression of CD40, CD80, CD86 IL-6 and IL-12p40 [63]. These results were also confirmed in intestinal tissue isolated from patients with ulcerative colitis, demonstrating that Chr expression was down-regulated in these patients compared to healthy controls [62]. Indeed, the mRNA levels of Chr were positively correlated with the mRNA expression of M2 macrophages activation markers and negatively to the expression of collagen, IL-8 and IL-18, but also with M1 activation markers (TLR-4 expression and NF-kB activation) and consequent pro-inflammatory cytokines production [61,62]. In these patients, the reduction of Chr level is also associated with a negative linear relationship with CD11c and CD86 [63]. Moreover, another study confirmed the anti-inflammatory effects of Chr on monocytes. Treatment with this peptide significantly inhibited the transcription of pro-inflammatory factors, such as NF-kB and AP-1, in these cells [64]. Furthermore, Rabbi and colleagues showed that Cts reduced intestinal inflammation and the onset of colitis lesions by a Stat-3 activation [65]. At the same time, the markers of M1 macrophage activation and the colonic levels of pro-inflammatory cytokines, such as IL-6, IL-1β and TNF-α, were significantly decreased by Cts treatment [66]. However, Cts did not influence M2 macrophage markers [66]. Furthermore, these anti-inflammatory effects were confirmed in cellular experiments using macrophages isolated from the peritoneal cavity and the bone marrow, demonstrating that in vitro treatment with Cts significantly decreased the production of the pro-inflammatory cytokines and phosphorylation of Stat-3 [65]. In addition, peritoneal macrophages isolated from naïve mice and treated with Cts and LPS displayed a reduction in the expression and production of pro-inflammatory cytokines blocking the activation of M1 macrophages [66]. In addition, the genetic deletion of Cts induces hypertensive conditions and left ventricular hypertrophy accompanied by significant macrophage infiltration in cardiac tissue and adrenal gland [67]. In this context, the absence of Cts induced an increased level of pro-inflammatory cytokines TNF-α, C-C motif chemokine ligand (CCL)-2, 3, C-X-C motif chemokine ligand (CXCL)-1 and catecholamines but also an elevated inflammation in heart with the up-regulation of cardiac genes, such as *Tnfa*, *Ifng*, *Emr1*, *Itgam*, *Itgax*, *Nos2a*, *IL12b*, *Ccl2* and *Cxcl1* [67]. It is of great interest that the intraperitoneal administration of Cts reversed this phenotype. In addition, macrophage depletion blocked the onset of hypertension in Cts-knockout (KO) mice [67]. Furthermore, bone-marrow transfer of KO animals in wild-type (WT) counterparts induced hypertension and cardiac inflammation, while opposite conditions showed the opposite phenotype [67]. All these data strongly suggest that the anti-hypertensive effects of Cts are partially mediated by an immunosuppressive action of this peptide on macrophages [67]. The role of Cts as an immunomodulator was also explored in the context of atherosclerosis and vascular injury. In fact, in vitro treatment with Cts on endothelial cells significantly reduces the release of TNF-α and vascular damage markers, such as ICAM-1 and VCAM-1, after LPS exposure [68]. At the same time, Cts treatment suppresses inflammatory responses and oxidizes the low-density lipoprotein-induced foam cell formation of human macrophages [68]. Kojima and colleagues demonstrated that Cts injection to ApoE^−/−^ mice significantly reduces macrophage infiltration and the consequent atherosclerotic lesions onset in the aorta but also suppresses aortic smooth muscle cells proliferation and collagen deposition in atheromatous plaques [68]. Furthermore, in vitro experiments with human aortic smooth muscle cells showed that Cts treatment can block collagen-1 and fibronectin expression and migration, proliferation and apoptotic process [68]. Of significant clinical impact, coronary artery disease patients displayed a substantial reduction of plasmatic levels of Cts but an increased expression in coronary atheromatous plaques [68]. In an acute pulmonary embolism in vivo model and cellular experiments with human pulmonary artery endothelial cells, Cts treatment was found to abolish thrombin-induced inflammation blocking TLR-4 expression and p38 phosphorylation, decreasing the consequent acute pulmonary embolism [69]. In conclusion, Vs-I and Chr than Cts are key attenuators of inflammation in different tissue and pathological conditions by reducing immune cell infiltration and inflammatory activation (Figure 2).

## 4. Dermaseptins and Anticancer Therapy

### 4.1. Dermaseptins

Dermaseptins (DRSs) are a class of peptides identified in the skin secretions of several Amazonian tree frogs of the family *Phyllomedusidae*, in particular, the species *Phyllomedusa* [70,71]. The first member of DRS family was isolated from the skin secretion of *P. sauvagei* and named DRS-S1 [72]. This 34-residues-containing peptide has antimicrobial activity against Gram-positive and Gram-negative bacteria, yeast and protozoa without affecting mammalian cells. The second is DRS-B2, isolated from exudates of *P. bicolor* [73,74,75]. It is also known as adenoregulin due to its capacity to increase the affinity of the agonist toward the receptor of adenosine A1 [76]. DRS-B2, with its 33 amino-acid residues, is considered the most abundant member and the most active peptide of the B family (B for the frog species *P. bicolor*). To date, more than 65 DRSs, listed in the APD3 database (https://aps.unmc.edu/, accessed on 1 June 2022) [26], have been isolated primarily from the skin secretion of South American tree frogs of the 67-member family *Phyllomedusidae* (https://amphibiansoftheworld.amnh.org/, accessed on 1 June 2022). Multiple alignments of the 67 sequences of DRSs listed in APD3 clearly showed that these polycationic peptides, rich in Lys residue, share a signature consisting of a conserved Trp residue at position three and a consensus AA(A/G)KAAL(G/N)A motif in the middle region [70]. Their MIC values are in the low micromolar range for a large panel of microorganisms, comprising bacteria (*S. aureus*, *E. coli*), yeast (*C. albicans*), filamentous fungi (*A. fumigatus*) and protozoa, such as *Leishmania Mexicana*, and show no hemolytic activity against human and rabbit erythrocytes. The mode of action by which DRSs kill these microorganisms follows the “carpet” mechanism [77,78]. These polycationic peptides, destructured in aqueous media, adopt an alpha helix structure upon contact with the host cell plasma membranes and then interact with their negative charges [79,80,81,82]. Once bound, the peptide will disrupt membrane permeability and cause the death of the microorganism (Figure 1).

### 4.2. Dermaseptins and Anticancer Properties

The first two anticancer DRSs peptides were isolated from the South American Amazonian tree frog, *Phyllomedusa bicolor*. These molecules, DRSs-B2 and DRS-B3, were tested in vitro against a human prostatic adenocarcinoma PC-3 cell line, showing an antiproliferative effect with an EC_50_ around 2–3 μM and demonstrating the inhibition of proliferation of more than 90% [83]. In addition, these two peptides also inhibited PC-3 cell colony formation in soft agar and the proliferation, differentiation and capillary formation of endothelial cells [83,84]. Furthermore, DRS-B2 blocks the proliferation and colony formation of several human tumor cell types, such as prostatic adenocarcinoma LNCAP, prostatic carcinoma DU145, mammary carcinoma (MDA-MB2318) cell lines and B-lymphoma lines [83]. These effects were also confirmed in vivo by a cell line PC3 murine xenograft model, showing that DRS-B2 inhibits tumor growth [83]. The anticancer mechanism of action of DRS-B2 was demonstrated by in vitro experiments with tumor PC3 cells. This peptide interacted with tumor cell surface, aggregating and penetrating the cells. Furthermore, it induced the release of cytosolic lactate dehydrogenase, a marker of cytotoxicity and necrosis, but no effects were observed on mitochondrial membrane potential and caspase 3 activations for apoptotic involvement [83]. Concerning the mechanisms of action of DRS-B2, confocal microscopy studies revealed that this peptide rapidly accumulates to cytoplasmic membranes, packed in vesicles and into the nucleus [85]. These effects were also partially mediated by glycosaminoglycans’ interaction with DRS-B2 and the consequent structural modification of the peptide with the α-helical domain [85]. Recently, a synthetic hormonotoxin molecule composed of dermaseptin-B2 associated with luteinizing hormone-releasing hormone (LHRH) was tested to improve the peptide’s antitumor activity, reducing its peripheral toxicity and lethality. This hormonotoxin displayed an anticancer effect very similar to DRS-B2 both in vitro and in vivo [86]. The LHRH addition to dermaseptin-B2 does not alter the peptide’s secondary structure and biological function [86]. On the other hand, double staining flow cytometry analysis showed that this hormonotoxin induced apoptosis instead of a necrotic process caused by DRS-B2 [86]. This different anticancer mechanism of action explains better tolerance and the lower toxicity of the hormonotoxin compared to dermaseptin-B2 [86]. In addition, other biochemical approaches have been used to increase the antitumor activity of DRS, delivering these agents in tumor cells, as seen in DRS-DStomo01 peptide [87]. DRS-DStomo01 was entrapped in chitosan nanoparticles, and the antitumor activity was tested in vitro against HeLa cells. The peptide induces DNA fragmentation and mitochondrial hyperpolarization with consequent cytotoxicity for cancer cells [87]. However, when used in chitosan nanoparticles, DRS-DStomo01 was more active than free peptides [87]. In 2016, two novel members of DRSs family were identified in the skin secretion of the frog *Pachymedusa dacnicolor* and called DRS-PD 1 and 2 [88]. Both peptides were reported to be active against many microorganisms, such as *E. coli*, *S. aureus*, *P. aeruginosa* and *C. albicans*, but with no lytic effects on mammalian red cells [88]. DRS-PD 2 displayed anti-proliferative effects against cancer cell lines, such as H157, PC-3 and U251-MG, within the concentration range of 10^−9^ to 10^−4^ M [88]. This property was also reported for DRS-PD 1 but only for human neuronal glioblastoma U251MG cell line [88]. In addition, these peptides could also inhibit the proliferation of human microvessel endothelial cells with the same concentration range for anticancer activity [88]. Other peptides from the South American orange-legged leaf frog (*Phyllomedusa hypochondrialis*) were identified and called DRS-PH. This peptide was active against several pathogens, such as *E. coli*, *P. aeruginosa*, *S. aureus* and its methicillin-resistant strain (MRSA), *E. faecalis* and *C. albicans*, in a concentration range from 1 μM to 512 μM [89]. Once again, this DRS displayed a broad spectrum of anticancer properties against different cancer cell lines, including MCF-7, H157, U251MG, MDA-MB-435S and PC-3 [89]. Very recently, different studies identified, from the skin of *Phyllomedusa sauvagei*, DRSs-PS type 1, 3 and 4, characterizing their anticancer properties [90,91,92]. In 2019, Long and coworkers demonstrated that DRS-PS1 has antimicrobial effects against *S. aureus*, *E. coli* and *C. albicans* [90]. Interestingly, DRS-PS 1 showed anti-proliferative effects on human glioblastoma U-251 MG, perturbing cell membrane integrity at the concentration of 0.1 μM [90]. Furthermore, the anticancer action with lower concentrations involves apoptosis activation by mitochondrial-related signal involvement [90]. DRS-PS 3 showed a broad spectrum of antimicrobial activities against several pathogens, such as *S. aureus*, *E. coli* and *C. albicans*, at high concentrations but with reduced cytotoxicity for erythrocytes [91]. However, the synthetic, more cationic and hydrophobic analogues created by replacing acidic amino acids D and E at 5 and 17, respectively, of the DRS-PS3 sequence by lysines (K5/D5, K17/E17-DRS-PS 3) or by replacing two neutral amino acids A10 and G11 with the hydrophobic amino acid leucine (L10/A10, L11/G11-DRS-PS 3) strongly increased their antimicrobial activities against the same pathogens with MIC values of 8 µM or less [91]. On the other hand, both artificial analogues exhibit a more significant hemolytic effect on red blood cells than DRS-PS 3 [91]. Furthermore, these peptides showed anticancer activities against H157, PC3 and HMEC-1 cell lines in the micromolar range but the most active was L10/A10, L11/G11-DRS-PS 3 [91]. Additionally, DRS-PS 4 displayed antimicrobial effects with many pathogens, such as *S. aureus* and MRSA, *E. faecalis*, *E. coli*, *P. aeruginosa* and *C. albicans*, in a range of concentrations from 1 µM to 32 µM, with biofilms eradicating properties of these microorganisms [92]. The antimicrobial mechanism of action is based on the ability of this peptide to permeabilize the bacterial cell membrane [92]. However, the hemolysis activity of DRS-PS 4 was tested using horse red blood cells showing slight effects at antimicrobial concentrations [92]. In addition, the anticancer activity of DRS-PS 4 was also evaluated on several human cell lines, including U251MG, MDA-MB-435S, H157, PC-3 and MCF-7, displaying a dose-dependent inhibitory activity with high cytotoxicity in a concentration range from 10^−9^ to 10^−4^ M [92]. On the other hand, it presents a slight suppressing effect on human microvascular endothelial cells [92]. Very recently, Dong et al. discovered the DRS-PP from frog *Phyllomedusa palliata*. This peptide was active at 2 µM against *E. coli*, *S. aureus* and MRSA, *C. albicans*, *P. aeruginosa*, *E. faecalis* and *K. pneumoniae* [93]. It is of great interest that DRS-PP showed anti-proliferative effects with cytotoxic activities on different cancer cells, such as H157, MCF-7, PC-3 and U251 MG, but no effects on human microvascular endothelial cells [93]. In vivo studies confirmed the anticancer property of this agent; in fact, DRS-PP was tested on a subcutaneous H157 tumor model of nude mice showing significant anti-tumor activity in a dose-dependent manner without hepatopulmonary and toxic side effects [93]. These effects are mediated via disruptive membrane action but exert pro-apoptotic effects induced by mitochondrial and death receptor pathways [93]. Finally, in 2021, DRS-TO was identified in the tiger-striped Leaf Frog, *Phyllomedesa tomopterna*, showing that this peptide was active against *S. aureus* and MRSA, *E. faecalis*, *E. coli*, and *C. albicans* [94]. Additionally, no hemolytic effect was observed on red blood cells, but DRS-TO showed anticancer activity against U251MG, H157 and PC-3 cancer cell lines at higher concentrations [94]. All these data report the great potential of DRSs as anticancer agents and their mechanism of action targeting membrane but also inducing pro-apoptotic effects by mitochondrial dysfunction and death receptor pathways (Figure 3).

## 5. Conclusions and Future Perspectives

In conclusion, the role of MF-AMPs is reported to be crucial in preventing infection. These molecules are essential for the first response to infections and may represent an alternative approach and open future clinical applications for the resolution of antibiotic resistance. At the same time, the endogenous MF-AMPs displayed different effects on different organs and cell types. Among these, they can influence the immune system’s inflammatory process and cellular components. In this perspective, CgA-derived peptides, such as Chr and Cts, are the perfect examples of immunomodulation mediated by MF-AMPs. On the other hand, exogenous MF-AMPs produced by different species but not in humans showed several potential therapeutic approaches. As reported in this review, DRSs appear to be excellent anti-proliferative factors with varying models of cancer cells. From a clinical point of view, these peptides may represent elective candidates for future anticancer therapy.

## Figures and Tables

**Figure 1 pharmaceutics-14-02014-f001:**
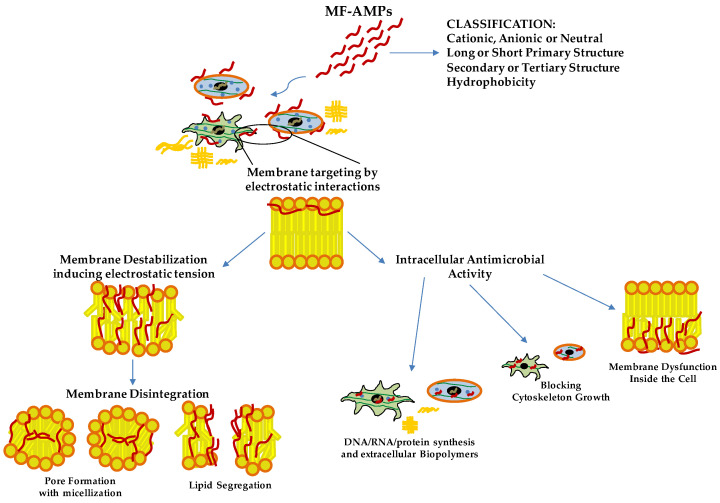
**Antimicrobial Mechanisms of Action of****Multifunctional Antimicrobial Peptides** (**MF-AMPs**). Based on their biochemical characteristics, MF-AMPs can interact with the membrane lipids, inducing the pathogens’ instability and rigidity of the cell membrane. Upon reaching the minimum inhibitory concentration (MIC) values, they cause pore formation and cell lysis. Some MF-AMPs exert a cell-penetrating peptides-like function. They can penetrate the membrane, blocking pathogens’ growth by internal membrane dysfunction, cytoskeleton interaction and interference of biomolecules production. Deoxyribonucleic acid (DNA); ribonucleic acid (RNA).

**Figure 2 pharmaceutics-14-02014-f002:**
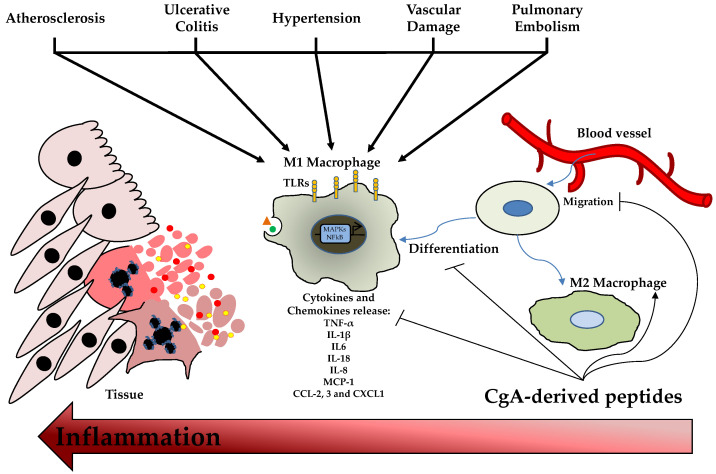
**ChromgraninA** (**CgA**)**-derived Peptides and Inflammation**. Damage or stress stimuli induce the migration, proliferation and activation of macrophages with consequent pro-inflammatory cytokines and chemokines release. This phenomenon generates cell death and inflammation. CgA-derived peptides can reduce M1 polarization and promote M2 anti-inflammatory macrophages differentiation. C-C motif chemokine ligand (CCL); C-X-C motif chemokine ligand (CXCL); interleukin (IL); monocyte chemoattractant protein (MCP); toll-like receptor (TLR); tumor necrosis factor (TNF).

**Figure 3 pharmaceutics-14-02014-f003:**
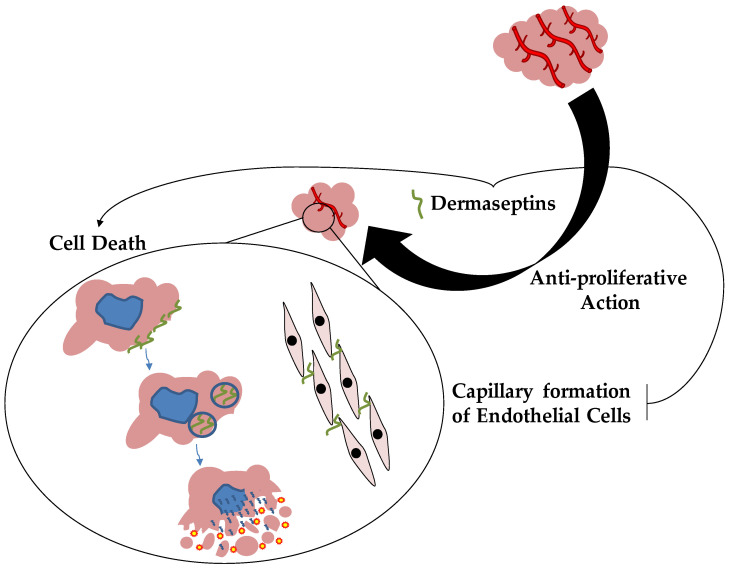
**Dermaseptins and Cancer**. Dermaseptins act as an anti-proliferative agent against several cancer cells in vitro and in vivo. The anticancer mechanism of action is based on the ability of these MF-AMPs to accumulate in cancer cells, inducing cell death and blocking tumor vascularization.

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
