# Peer review of "Recent Advances in Multifunctional Antimicrobial Peptides as Immunomodulatory and Anticancer Therapy: Chromogranin A-Derived Peptides and Dermaseptins as Endogenous versus Exogenous Actors"

_pharmaceutics, 2022, doi:10.3390/pharmaceutics14102014_

Round 1

Reviewer 1 Report

The manuscript analyzes the literary data on the activity of two classes of antibacterial peptides - chromogranin A derivatives and dermaseptines. Moreover, the former are considered as endogenous antibacterial peptides (AMPs), and the latter as exogenous. So the first question is, why did the authors focus on these AMP classes? What is the main difference in the activity of AMPs belonging to these classes? The use of which AMPs is most promising for medical purposes?

Some minor questions:
Lines 76-77: "they are not toxic at high concentrations, and most of these are modulators of an immune response without generating pathogen resistance"
The toxicity of AMPs to eukaryotic cells is the main limitation in their clinical use. Immune response modulation and antibacterial activity occur in different concentration ranges. Microbial resistance to AMP (direct antimicrobial action) and immune response to microorganisms are two different processes.

Lines 78-81: "they are not toxic for host cells due to the specific interaction with microorganism membranes, i.e., membrane compounds, such as lipids or specific microbial ligands for antimicrobial peptides, hydrophobicity, transmembrane, and charge potential"
What specific interaction of AMPs with bacterial membranes are we talking about?

Lines 93-94: "Some MF-AMPs, classified as cell-penetrating peptides"
CPP is a separate, self-contained class of peptides which, for the most part, do not possess antimicrobial properties.

Lines 98-99: "membrane receptor-mediated"
AMP has its own receptors on bacterial membranes?

Lines 109-111:  "some AMPs do not degrade the membrane but may translocate across it. These peptides are called cell-penetrating peptides (CPPs)".

CPP is a separate, self-contained class of peptides which, for the most part, do not possess antimicrobial properties.

Author Response

The authors thank reviewer 1 for his relevant comments and questions. We have modified the text according to the comments of reviewer 1.

Reviewer 1

The manuscript analyzes the literary data on the activity of two classes of antibacterial peptides - chromogranin A derivatives and Dermaseptines. Moreover, the former are considered as endogenous antibacterial peptides (AMPs), and the latter as exogenous.

So the first question is, why did the authors focus on these AMP classes? What is the main difference in the activity of AMPs belonging to these classes? The use of which AMPs is most promising for medical purposes?

We focused on these two classes of AMPs because both have great potential for clinical applications. In addition, Dr. M-H M-B, to which this special issue is dedicated, has long experience with Chromogranin-derived peptides. They are reported as AMPs but also classified as multifunctional AMPs (MF-AMPs). We have discussed these two classes due to the main biomedical applications as an alternative to antimicrobial action, i.e., immunomodulation and activity against cancer. Furthermore, these peptides act with a similar mechanism of antimicrobial action, but their difference exists precisely in the figure of MF-AMPs. Chromogranin A-derived peptides act as a modulator of the immune response, while Dermaseptins as powerful anti-cancer agent. We also think these different activities are justified because one class of peptides is endogenous while the other is exogenous. Thus, we focused on these peptides because 1. they are elective examples of MF-AMPs and 2. for their relevant clinical potential in immunomodulation and therapy against cancer.

All this information is briefly summarized at the end of the introduction

Some minor questions:

Lines 76-77: "they are not toxic at high concentrations, and most of these are modulators of an immune response without generating pathogen resistance"

The toxicity of AMPs to eukaryotic cells is the main limitation in their clinical use. Immune response modulation and antibacterial activity occur in different concentration ranges. Microbial resistance to AMP (direct antimicrobial action) and immune response to microorganisms are two different processes.

We thank the Reviewer for the relevant comment and apologize for not being clear enough. Concerning toxicity, we alluded to the endogenous peptides produced by the host organism reported as not toxic for it.

For the two activities reported, we know that they are different, but the peculiarity of the MF-AMPs, the object of the review, is to discuss other functions, albeit at different concentrations.

We revised the text in new Line 76-77 to clarify our idea concerning this sentence:

'In addition, the endogenous AMPs are not toxic at high concentrations, and some of these are reported as multifunctional agents with immunomodulatory activity without generating pathogen resistance.'

Lines 78-81: "they are not toxic for host cells due to the specific interaction with microorganism membranes, i.e., membrane compounds, such as lipids or specific microbial ligands for antimicrobial peptides, hydrophobicity, transmembrane, and charge potential."

What specific interaction of AMPs with bacterial membranes are we talking about?

We modified that section and introduced the notion of “electrostatic interactions”:

Lines 78-81: "they are not toxic for host cells due to the specific electrostatic interaction with microorganism membranes, microbial membrane compounds, such as negatively charged lipids or specific microbial ligands for antimicrobial peptides. hydrophobicity, transmembrane, and charge potential."

Lines 93-94: "Some MF-AMPs, classified as cell-penetrating peptides."

CPP is a separate, self-contained class of peptides which, for the most part, do not possess antimicrobial properties.

The authors agree with the comment of the Reviewer. However, some MF-AMPs (i.e., CgA-derived peptides (Zhang et al., 2009. Doi: 10.1371/journal.pone.0004501)) show the activity related to CPP. However, this activity is reported in relation to the interactions with host cells. Lines 93-94 have been modified, replacing CPP with a more general citation on cell-penetrating peptides-like function:

Lines 93-94: " Some MF-AMPs exert a cell-penetrating peptides-like function. They…"

Related to that point, we have modified Figure 1 (reporting the general mechanism of action of AMPS), deleting Cell Penetration Peptide and clarifying the figure.

Lines 98-99: "membrane receptor-mediated"

AMP has its own receptors on bacterial membranes?

We apologize for not being clear enough. We revised the text in new Line 102 to clarify this sentence:

based on electrostatic interactions with lipid and membrane receptor ligands-mediated

Lines 109-111: "some AMPs do not degrade the membrane but may translocate across it. These peptides are called cell-penetrating peptides (CPPs)".

CPP is a separate, self-contained class of peptides which, for the most part, do not possess antimicrobial properties.

As previously described, new line 117 has been modified, replacing CPP with a more general citation on cell-penetrating peptides-like function:

These peptides exert a cell-penetrating peptides (CPPs)-like function are called cell-penetrating peptides (CPPs).

Reviewer 2 Report

Overall an informative review. I must say that the first two sections were rather difficult to follow. Here I suggest a rather careful editing of the English language and style (I give a non-comprehensive list of suggestions/corrections below). Sections 3 and 4, on the other hand, are well-written and informative. On these latter sections I have no specific comment.

Finally, I found the review a bit short (but I must say that I do not follow AMPs literature attentively), however the number of citations in sections 3 and 4 seem appropriate for the type of manuscript. 

Overall, considering the above, in my opinion the manuscript can be considered for publication in the journal, provided that sections 1 and 2 are carefully edited.

Lines 36-38: the word/definition "system" appears three times in close succession. Please consider revising.

Lines 42-43: References are needed for "different antimicrobial mechanisms of action" and "different intensities against an extensive collection of pathogens."

Lines 44-48: Rather long sentence (and a bit hard to follow). Please revise.

Line 53: "The 20th century will be reminded of when antibiotic..." I would suggest: "The 20th century will be remembered as the time when antibiotic..." or similar

Line 115: "avoid" perhaps "prevent"?

Line 123: just as an example (this type of construction appears several times in the manuscript): "Different biomaterials functionalized peptides and peptidomimetics agents" sounds better if written "Different peptides and peptidomimetics agents functionalized biomaterials" or closer to original: "Different biomaterials functionalized with peptides and peptidomimetics agents". Here I assume that the authors mean that it is the biometerial that is functionalized, not the peptide. Or perhaps I am not really catching the meaning?

Line 156: "strong interaction" maybe better than "solid interaction"

Author Response

Reviewer 2

Overall an informative review. I must say that the first two sections were rather difficult to follow. Here I suggest a rather careful editing of the English language and style (I give a non-comprehensive list of suggestions/corrections below). Sections 3 and 4, on the other hand, are well-written and informative. On these latter sections I have no specific comment.

Finally, I found the review a bit short (but I must say that I do not follow AMPs literature attentively), however the number of citations in sections 3 and 4 seem appropriate for the type of manuscript.

Overall, considering the above, in my opinion the manuscript can be considered for publication in the journal, provided that sections 1 and 2 are carefully edited.

We thank the Reviewer for this suggestion, and accordingly, we have modified the text providing an extensive revision of sections 1 and 2.

Lines 36-38: the word/definition "system" appears three times in close succession. Please consider revising.

We have edited the text:

'It consists of different molecules produced and released by various cell types belonging to the organs, immune or other systems [1]. Among these molecules, we find proteins with direct antimicrobial activity or which activate the complement system proteins.'

Lines 42-43: References are needed for "different antimicrobial mechanisms of action" and "different intensities against an extensive collection of pathogens."

We added references in the text. In particular for:

different antimicrobial mechanisms of action (Riera Romo et al., Immunology 2016. doi: 10.1111/imm.12597; Auvynet, et al., FEBS J 2009. doi: 10.1111/j.1742-4658.2009.07360.x)

different intensities against an extensive collection of pathogens (Riera Romo et al., Immunology 2016. doi: 10.1111/imm.12597; Auvynet, et al., FEBS J 2009. doi: 10.1111/j.1742-4658.2009.07360.x; Wang et al Nucleic Acids Res 2004. doi: 10.1093/nar/gkh025; Brandwein et al., Front Immunol 2017. doi: 10.3389/fimmu.2017.01637)

Lines 44-48: Rather long sentence (and a bit hard to follow). Please revise.

We have revised the long sentence by shortening the length according to the Reviewer's suggestions. In particular, we split into two sentences:

However, in light of the potential human clinical application, they can be classified as endogenous AMPs, produced by the human organism, such as Defensins, Cathelicidins and Dermcidins [3,4]. On the other hand, exogenous AMPs are produced by microorganisms, plants, insects, amphibians and fishes or mammals but not identified in humans, such as Thionins, Piscidins, Cecropins and Dermaseptins [5-9].

In addition, we revised all sections 1 and 2 in this light.

Line 53: "The 20th century will be reminded of when antibiotic..." I would suggest: "The 20th century will be remembered as the time when antibiotic..." or similar

The authors agree with the comment of the Reviewer. We replaced the sentence as suggested by reviewer 2.

Line 115: "avoid" perhaps "prevent"?

We agree with the comment of the Reviewer. Thus, we have replaced the word in the new line 120, as suggested by reviewer 2.

Line 123: just as an example (this type of construction appears several times in the manuscript): "Different biomaterials functionalized peptides and peptidomimetics agents" sounds better if written "Different peptides and peptidomimetics agents functionalized biomaterials" or closer to the original: "Different biomaterials functionalized with peptides and peptidomimetics agents."

We apologize for not being clear enough. Yes, we mean that peptides functionalize the biomaterial. Thus we revised the text in the new line 128 with this sentence:

'Different biomaterials functionalized with peptides and peptidomimetics agents are available for clinical applications.'

Line 156: "strong interaction" may be better than "solid interaction"

The authors agree with the comment of the Reviewer. We replaced strong with solid in the t
